# The Revealed Mechanism of Rock Burst Based on an Innovative Calculation Method of Rock Mass Released Energy

**DOI:** 10.3390/ijerph192416636

**Published:** 2022-12-11

**Authors:** Wenlong Zhang, Jicheng Feng, Ji Ma, Jianjun Shi

**Affiliations:** 1School of Civil Engineering and Architecture, Qingdao Huanghai University, Qingdao 266427, China; 2School of Safety Engineering, North China Institute of Science & Technology, Langfang 065201, China; 3College of Safety Science and Engineering, Henan Polytechnic University, Jiaozuo 454003, China

**Keywords:** released energy, calculation method, numerical simulation, rock mass, rock burst

## Abstract

It is very necessary to study the mechanism of rock burst, which is related to the safe construction of many geotechnical projects. Previous studies have shown that small trigger stress will lead to large energy release, but the specific conditions that cause the release and how to quantify the energy are urgent problems to be solved. In this study, an innovative calculation method of rock mass energy release is proposed, and the calculated release energy is consistent with the monitoring results of field monitoring equipment. The revealed mechanism of rock burst reflected is that under the condition of a large-ratio pre-state stress field (mostly > 2.5), a small trigger stress field will lead to a large amount of energy release under “late butterfly shape” or “final butterfly shape” of the plastic zone. This study reveals the key factor of rock burst, which plays an important reference role for the mechanism research, subsequent monitoring and treatment method of rock burst.

## 1. Introduction

Rock bursts are harmful phenomena in many geotechnical engineering fields [1,2], such as rock bursts in tunnels [3], rock bursts in metal mines [4], rock bursts in coal mines [5], coal and gas outbursts [6], and dynamic responses of metro engineering [7]. These rock bursts are obviously different, but they also have common parts. They all occur in rocks or soil and other media and are caused by certain forces and structures. Rock burst have brought great losses to mankind which can cause instantaneous damage to tunnels, mine roadways, etc., and bring a great threat to safe production [8].

At present, research on rock bursts mainly focuses on their occurrence mechanism [9], monitoring methods [10,11], early warning indicators [10] and treatment measures [9,12]. Among the above, the occurrence mechanisms should be studied clearly first, to form the basis of follow-up monitoring, early warning and treatment measures. There are many studies on the mechanism of rock bursts, which mainly include energy theories [13], strength theories [14], stiffness theories [15], rock burst tendency theories [16], instability theories [17], three factor theories [18], three criteria theories [19], dynamic and static load principles [20], rock burst initiation theories [21], butterfly rock burst mechanism [22,23], etc. No matter which mechanism, it must be aimed at the rock mass involved in the disaster, the mechanical properties of the rock mass, the stress environment leading to the disaster, the time process of the disaster, etc.

In terms of rock burst research methods, some mechanism studies aimed at a specific condition, such as fault [24,25], strong mining influence [26,27], high stress environment [28], high dynamic load [29], stratum movement [30], etc. Others study a series of problems from one aspect, such as strength [31], stress [32], energy [33], gas superposition [34], surrounding rock properties [35,36,37] and so on [18]. Whether it is into specific conditions or a certain aspect, the research plays a positive role in revealing the rock burst mechanism. However, research into common problems of geotechnical engineering is more inclined to use the general mechanism to explain different manifestations, which is the highest goal of scientific and engineering research.

As for rock burst, an important point of view is that small inducing factors lead to a serious dynamic response [38]. However, there are two main problems: one is the specific condition that small inducing factors produce a large energy release; the other is the quantifiable degree of released energy, that is, how to calculate the released energy and match it with the energy of an on-site vibration signal monitoring system. This study attempts to reflect the mechanism of rock bursts through the simplest mechanical calculation method and the simplest model. An innovative calculation method of energy released from a rock mass is proposed from the perspective of energy in this study. With the help of a numerical simulation method, the energy-release law and specific value of rock burst are calculated, and the formation factors and mechanism of rock burst are obtained. The mechanism research of rock burst mostly depends on the numerical simulation method with strong operability. In addition to the mechanism of rock burst, this study also discusses the energy-release laws under different circumstances, such as no holes, uniform stress field, non-uniform medium, etc. The study is of great significance for revealing the mechanism of rock bursts and provides a basis for subsequent monitoring and treatment methods of rock burst.

## 2. Numerical Methodology

The surrounding rock in geotechnical engineering is basically in the state of three-dimensional stress. The rock mass in a certain area is taken as the study object, and its volume is assumed to be Ω. A circular hole is set in the middle of the Ω area in order to facilitate the research (in respect of tunnels or roadways). The triaxial force of the Ω rock mass is simplified to the form of (P_1_, P_2_, P_3_), where P_1_ is the maximum force, P_2_ is the intermediate force, and P_3_ is the minimum force. Under the action of external force, each element named as (f(x, y, z) in the rock mass will produce a unit stress, which is assumed to be (σ_1i_, σ_2i_, σ_3i_), where σ_1i_ is the maximum principal stress, σ_2i_ is the intermediate principal stress, and σ_3i_ is the principal minimum stress. Under the action of stress, the element will produce a certain energy, which has been clearly pointed out as Equation (1) in previous research results [31].
(1)$$f\left(x,\;y,\;z\right)=\frac{1}{{2E}_{i}}\left[\sigma_{1i}{}^{2}{+\sigma }_{2i}{}^{2}{+\sigma }_{3i}{}^{2}{-2\mu }_{i}\left(\sigma_{1i}\sigma_{2i}{+\sigma }_{2i}\sigma_{3i}{+\sigma }_{1i}\sigma_{3i}\right)\right]$$
where E_i_ is the elastic modulus of the element, and μ_i_ is the Poisson’s ratio of the element.

Under the action of a certain trigger stress field (TSF, the stress wave caused by roof fracture [39] or media fracture [40]), the force state and energy value of the rock mass will change. It is assumed that the state of rock mass before TSF influence is a pre-state stress field (PSSF), and the state after TSF influence is a late-state stress field (LSSF). Assuming the rock mass as two kinds of medium, one is pure elastic and the other is an elastoplastic medium. Here, we define the storage energy of pure elastic medium and elastoplastic medium under PSSF as U_PSSF_ and U_PSSF_^′^, respectively, and similarly define the storage energy of pure elastic medium and elastoplastic medium under LSSF as U_LSSF_ and U_LSSF_^′^, respectively. Under the condition of an elastic medium, all elements in the rock mass are elastic elements, while under the condition of an elastoplastic medium, some elements in the rock mass are elastic elements (Ω_e_) and the other are elastoplastic elements (Ω_p_) due to the plastic failure. When the mechanical state changes from PSSF to LSSF, new plastic elements of ΔΩ_p_ are generated, the original Ω_e_ changes to Ω_p_−ΔΩ_p_, and the original Ω_p_ changes to Ω_p_ + ΔΩ_p_. Therefore, U_PSSF_, U_PSSF′_, U_LSSF_, and U_LSSF′_ can be expressed by Equations (2)–(5), respectively.
(2)$$U_{\mathrm{FSF}}=\int \int \int_{\Omega (\mathrm{PSSF})}^{}f(x,y,z)\mathrm{dV}\;$$
(3)$$U_{\mathrm{FSF}}{}^{\prime }=\int \int \int_{\Omega_{e}\left(\mathrm{PSSF}\right)}^{}{f(x,y,z)\mathrm{dV}}_{e}+\int \int \int_{\Omega_{p}\left(\mathrm{PSSF}\right)}^{}{f(x,y,z)\mathrm{dV}}_{p}$$
(4)$$U_{\mathrm{LSF}}=\int \int \int_{\Omega (\mathrm{LSSF})}^{}f(x,y,z)\mathrm{dV}$$
(5)$$U_{\mathrm{LSF}}{}^{\prime }=\int \int \int_{{(\Omega }_{e}-\Delta \Omega p)(\mathrm{LSSF})}^{}{f(x,y,z)d(V}_{e}{-\Delta V}_{P})+\int \int \int_{{(\Omega }_{p}+\Delta \Omega p)(\mathrm{LSSF})}^{}{f(x,y,z)d(V}_{p}{+\Delta V}_{P})$$
where V is the volume of all elements, and V_e_ and V_p_ are the volume of elastic elements and elastoplastic elements, respectively.

The energy different-value D_PSSF_ represents the energy difference between U_PSSF_ and U_PSSF′_ under PSSF state, and similarly the energy different-value D_LSSF_ represents the energy difference between U_LSSF_ and U_LSSF_^′^ under the LSSF state. The energy difference of the different-value of the two-forces state represents the release energy from the PSSF state to LSSF state, as shown in equation 6 (only part of the total energy is converted into elastic wave energy [41], so multiply by transformation factor β, which is generally between 1% and 10% [26,42]). When a rock burst occurs, the energy value reflected by vibration signals monitored by vibration signal monitoring system is W.
(6)$${W=\beta ((U}_{\mathrm{LSSF}}{-U}_{\mathrm{LSSF}}{}^{\prime }){-(U}_{\mathrm{FSSF}}{-U}_{\mathrm{FSSF}}{}^{\prime }))$$

The flow chart of the innovative energy calculation method established above is shown in Figure 1. Firstly, the energy different-value between elastic medium and elastoplastic medium is calculated under the corresponding mechanical states, and then the difference between the energy of the two mechanical states to obtain the specific energy value released in the process of mechanical state change is calculated. The energy different-value between the two mechanical states is not a simple subtraction of the energy under the condition of elastoplastic medium, because the elastic medium energy of the foundation is different under different mechanical states. The innovation of this method is that the energy release of two mechanical states before and after TSF is obtained by four models subtracted in pairs. The elastic wave energy obtained by this method can be compared with the actual energy obtained by vibration signal acquisition when a rock burst occurs, and then the occurrence factors of rock burst can be obtained. In addition, the sizes of the four models are consistent, and the influence of model size is eliminated by subtracting two by two. It should be pointed out that the three-dimensional mechanical state of the actual element on site is difficult to obtain through actual measurement methods; however, the mechanical state of each unit can be obtained by a numerical simulation method. Therefore, it is best to complete energy calculation and mechanism disclosure with the help of numerical simulation tools. The obtained model and calculation method quantify the capacity value before and after the occurrence of a rock burst, which is of great help to the study of the rock burst mechanism in this study. In addition, it can also be used to calculate the damage or accumulation of rock mass energy.

## 3. Results

According to the methodology proposed above, the simplest model and the stress state are established to calculate the energy, as shown in Figure 2. The model size is 200 × 200 × 1 m, corresponding to the width, height and thickness of the model, respectively. The PSSF assigned to the model is (P_1_, P_2_, P_3_), and the application directions are up and down, front and back, left and right, respectively. A small hole with a diameter of 5.6 m is set in the middle of the model to represent the tunnel or roadway (5.6 m represents the specific diameter of a field test in a roadway, which is applied to the research model). The medium used is a kind of coal (rock burst accidents occur more often in coal, so selecting coal as the medium can better reflect the real mechanism), with shear strength of 1.3 GPa, cohesion stress of 3 MPa, friction angle of 25°, compressive strength of 15 MPa and tensile strength of 1.77 MPa. The initial PSSF is set to (20, 20, 20) MPa, and the TSF is set to 1 MPa and only increases to P_1_ each time for the convenience of calculation. When calculating the stress and energy value of an elastic medium, the command of “model mech elastic” in FLAC^3D^ is adopted, and when calculating the elastoplastic medium, the “model Mohr” in FLAC^3D^ is adopted (the Mohr coulomb criterion is adopted as the failure criterion).

After dozens of numerical simulations and energy calculations, the plastic zone distribution and energy values under different mechanical states are obtained. Results show that the distribution of the plastic zone of the elastoplastic medium experienced a process of “non-butterfly–early butterfly–final butterfly”, and the “early butterfly” is defined when P_1_ is 50 MPa (η = P_1_/P_3_ = 2.5, which is similar to [43]), and the “final butterfly” is defined when P_1_ is 55 MPa (η = P_1_/P_3_ = 2.75). The results of plastic zone distribution and energy distribution of the elastoplastic medium show that the plastic zone and energy form an obvious concentration around the hole, and the distribution range and concentration degree expand with the increase of P_1_, which is confirmed in Figure 3. The energy different-value in most areas is positive, but there are also some areas with a negative value, indicating that the energy of the elastic medium is less than that of the elastoplastic medium for the energy of a small part of elements. It can also be concluded from the energy difference value distribution that the distribution result of energy is obviously related to the distribution result of the plastic zone, and the area producing the plastic zone seems to have a greater energy difference.

The variation curve of energy released is shown in Figure 4. The release of energy is divided into three stages: “pregnant period”, “growth period” and “upward period”. The dividing points of the three stages are also “early butterfly shape”, “late butterfly shape” and “final butterfly shape”, and the corresponding P_1_ values of LSSF are 50 MPa, 55 MPa and 58.6 MPa respectively. The above facts show that even if the TSF is the same 1 MPa, the effect of the initial PSSF on the release of energy is obviously different (the amount of energy released increases by more than 10 times, and in the “upward period” stage, the amount reaches even 10^8^ J).

From the above results, the basic characteristics of plastic zone distribution and energy distribution of the system under different PSSF conditions are obtained. The corresponding relationship between released energy and plastic zone is obvious. In the “growth period” and “upward period” stages, the released energy caused by the same TSF are significantly increased, which is very consistent with the occurrence mechanism of rock burst. That is, a small TSF can lead to a large energy release, especially under the condition of a large ratio PSSF (deviatoric loading [44], mostly η > 2.5). In the “pregnant period”, the TSF required to reach the accident energy value will be very large; however, in the “growth period” and “upward period” stages, the TSF required will be significantly smaller.

For the amount of released energy, the above results show that in the “growth period” and “upward period” stages, the total energy release of one meter of roadway or tunnel is about 10^7^–10^8^ J. Considering the elastic wave energy transformation factor β (1~10%), the roadway of 10~100 m length will release 10^7^~10^8^ J of elastic wave energy, which is consistent with the actual released energy monitored by the on-site vibration signal monitoring system. Therefore, according to the innovative calculation method and its calculation results, the factors of a rock burst are reflected in the model: hole; P_1_ greater than uniaxial compressive strength (reaching the value of foundation failure); PSSF with higher η (mostly η > 2.5) and an appropriate TSF. Among the factors, hole is necessary for excavation and production; P_1_ is greater than uniaxial compressive in the “growth period” and “upward period” in the high probability; the TSF reaching the energy critical value is small (Ref. [45] shows that it mostly much less than 1 MPa); therefore, the PSSF with a higher η may be the key factor of a rock burst. The mechanism of rock burst reflected in this study is that under the condition of a large-ratio PSSF (mostly η > 2.5), a small TSF will lead to a large amount of energy release under the “late butterfly shape” or “final butterfly shape” of the plastic zone.

## 4. Discussion

The point to prove the key influence of a large-ratio PSSF is whether there is a large increase in energy under a uniform stress field applied to the model; therefore, Figure 5 and Figure 6 are made. From the results, when P_1_, P_2_ and P_3_ are increased by 5 MPa at the same time, the storage energy of the pure elastic medium and the elastoplastic medium are also increasing; however, the different value of energy does not increase abruptly. Even if the PSSF is bigger than (55, 55, 55) MPa and the TSF is bigger than (5, 5, 5) MPa, the released energy of the system is still only at the 10^7^ J level. In addition, there is no butterfly shape in the plastic zone under PSSF = (P_1_, P_2_, P_3_) = (55, 55, 55) MPa, which also confirms the key role of the butterfly plastic zone caused by a large-ratio PSSF.

One of the formation factors of a rock burst is the existence of a hole. Will there be a sudden change in energy in the model without a hole? In order to show this, Figure 7 and Figure 8 are made to show a change curve of storage energy, different-value, and released energy of the no-hole model. The results show that the storage energy of the pure elastic medium and elastoplastic medium also increase, but there is no difference before the final failure, resulting in a different-value of 0; however, when the failure occurs, there is a large energy different-value. The released energy is 0 before failure occurs and increases sharply to 10^8^ J after failure occurs. From the shape of the plastic zone, partial failure occurs after failure, and complete failure occurs when P_1_ reaches 59 MPa. The above facts show that the mutation of energy is due to the failure of the model, and the existence of a hole will lead to stress concentration [46] in the model in advance, and the guess is that the larger the diameter, the greater the impact, which will be studied in detail in the later study.

The above calculation and simulation results are based on a uniform medium, and whether the proposed factors and mechanisms are applicable to non-uniform medium [47,48], especially a shallow layered medium, is worth verifying. Therefore, the numerical simulation model in Figure 9 is established. The established model is divided into five layers; the middle layer is coal, with the lowest strength and a thickness of 11 m, and a hole is located in the coal seam. The first layer in the upper part is mudstone with a thickness of 26 m, the second layer in the upper part is conglomerate with a thickness of 21 m; the first layer in the lower part is mudstone with a thickness of 6 m, and the second layer in the lower part is fine sandstone with a thickness of 16 m. The total height and width of the model are all 80 m, and the external force setting is the same as that of the uniform medium model.

Figure 10 and Figure 11 denote the change curve of storage energy, different-value, and released energy of a non-uniform medium. Results show that both the storage energy of the pure elastic medium and the elastoplastic medium increase with the increase of P_1_, and the increasing speed is also different, which is the same as the uniform medium. The different-value of energy also increases for the non-uniform medium model; however, the acceleration is smaller compared to a uniform medium. From the results of the energy release, the non-uniform medium also has a sudden change, and the dividing points are also distributed at η = 2.5, 2.75 and 2.93. For the distribution of the plastic zone, the difference is that the plastic zone entirely extends to the model boundary along the direction of the weak coal seam when P_1_ = 60 MPa. The above facts show that the proposed energy calculation method and mechanism are also suitable for a ubiquitous non-uniform medium.

This study reveals the important impact of a large-ratio PPSF on a rock burst disaster. In fact, some previous examples [49,50] also show that many rock burst accidents occur in places with large stress ratios. Some of these places are affected by faults and others are in a large abnormal stress field environment. The research results are consistent with the rock burst accident occurring when heading to the corresponding area, while still occurring when mining the same area, and the area is more likely to be in the environment of a large-ratio stress field. When heading and mining to this area, a small TSF will easily lead to the rock burst accident. In practice, a TSF is unlikely to be the dominant factor causing accidents, unless a hard roof hundreds of meters thick is broken on a large scale at one time (TSF above 1 MPa is required in the non-butterfly state) [51]; however, this rarely exists in reality, which has been already demonstrated in [45].

The energy variation law and mechanism obtained by the study are based on an innovative calculation method and numerical simulation method. Results show that a large-ratio PSSF and butterfly plastic zone and their energy mutation are closely related to rock burst accidents. The study points out the direction for later monitoring and early warning of rock burst, that is, the state of a large-ratio PSSF or plastic zone. Once the stress ratio reaches 2.5/2.75/2.93, or the butterfly state of plastic zone has formed (like the results in [52]), the state must be more dangerous. It should be recognized that the element stress values in this study are obtained by numerical simulation (in the energy calculation model, the energy calculation parameters of the element in the plastic zone adopt the results derived from FLAC^3D^, and adopt the energy calculation method equivalent to the elastic element, which will bring some errors, but will not change the law of energy release), and it is difficult to monitor the three-dimensional stress state of every element in the actual field to verify the correctness of the method. The next step is likely to be reflected in the laboratory or by field verification (such as the results in [53]) of the method, and seek monitoring methods to monitor the three-dimensional stress field or real-time plastic zone state.

## 5. Conclusions

In this study, an innovative calculation method of rock mass release energy is established, and a rock burst mechanism based on energy analysis and plastic zone distribution is obtained through the calculation method. The main conclusions are as follows: (1) The energy release area of a rock mass is closely related to the distribution area of the plastic zone. In the butterfly plastic zone stage, the energy release of rock mass increases significantly. (2) For the rock mass with a butterfly plastic zone, a small TSF can lead to a large amount of energy release, resulting in the occurrence of a rock burst. (3) A butterfly plastic zone is caused by a large-ratio PSSF (η = 2.5, 2.75 and 2.93), which is the key factor leading to a rock burst. However, it should be pointed out that this study relies more on numerical simulation, and later research should focus more on laboratory or field verification.

## Figures and Tables

**Figure 1 ijerph-19-16636-f001:**
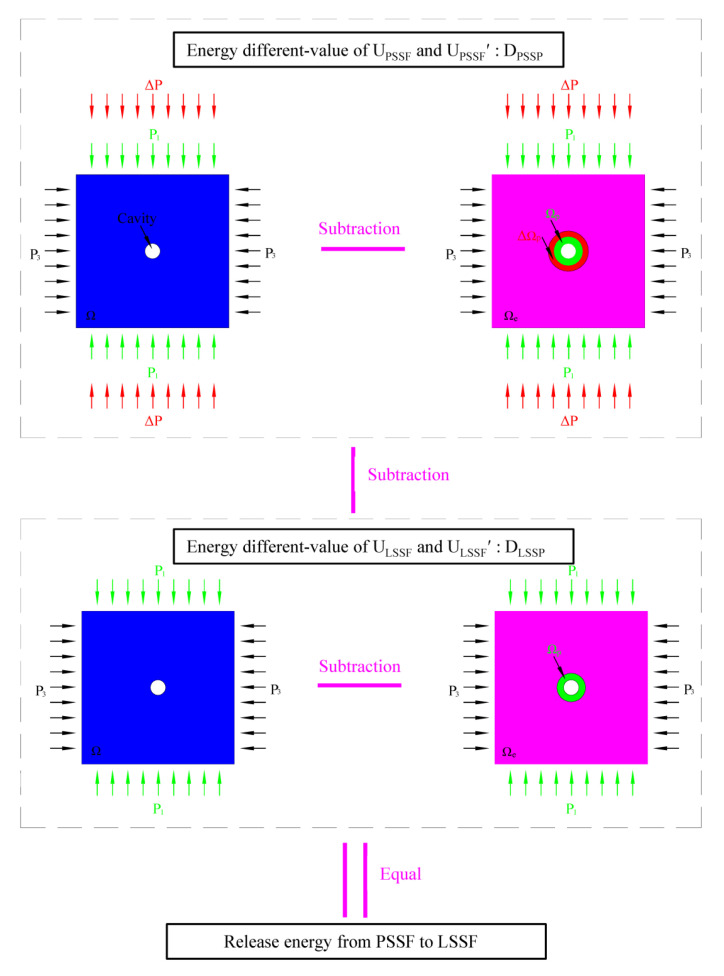
Flow chart of the innovative energy calculation method.

**Figure 2 ijerph-19-16636-f002:**
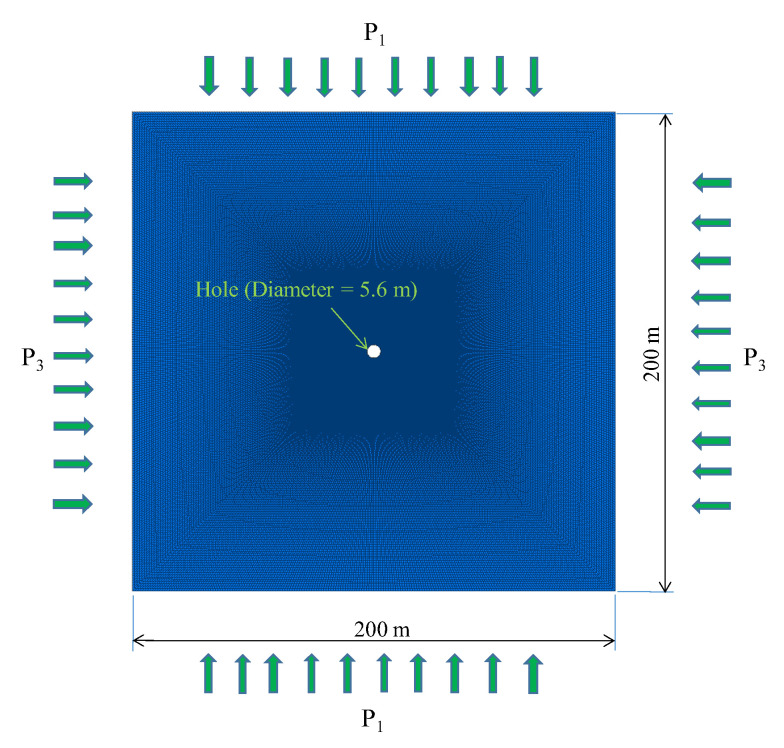
Model and force state to calculate the energy.

**Figure 3 ijerph-19-16636-f003:**
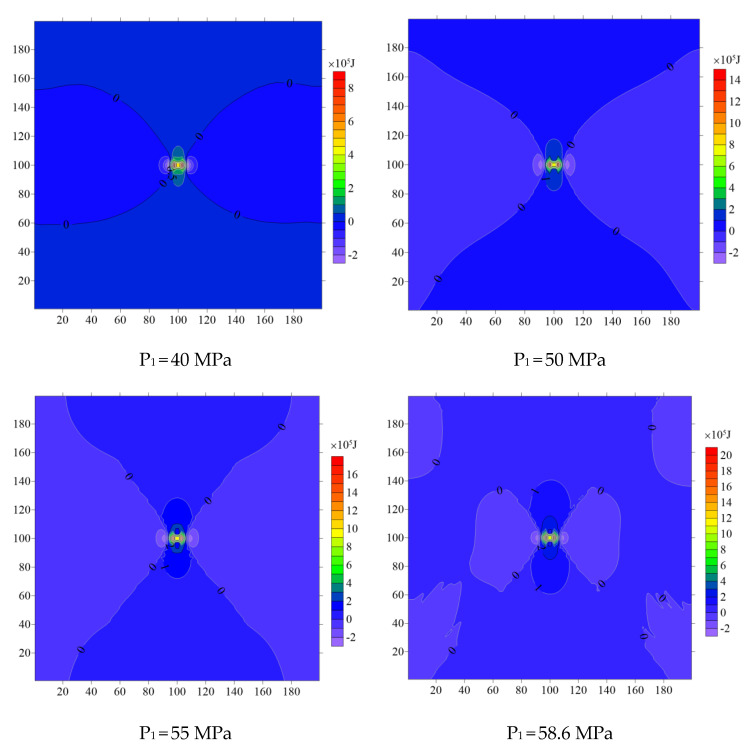
Energy different-value distribution of typical mechanical states.

**Figure 4 ijerph-19-16636-f004:**
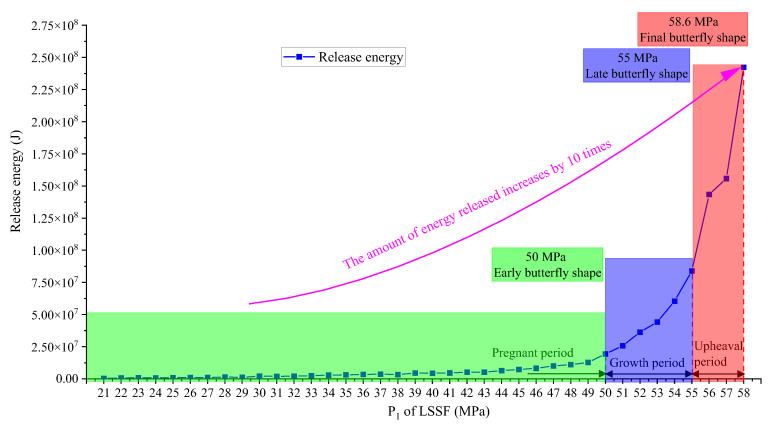
Change curve of release energy.

**Figure 5 ijerph-19-16636-f005:**
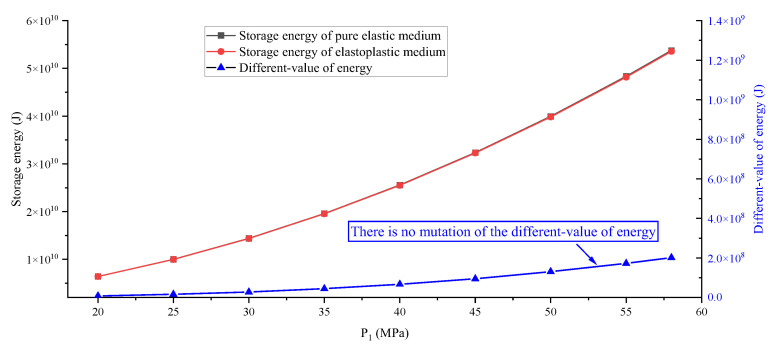
Change curve of storage and different-value of energy under a uniform stress field.

**Figure 6 ijerph-19-16636-f006:**
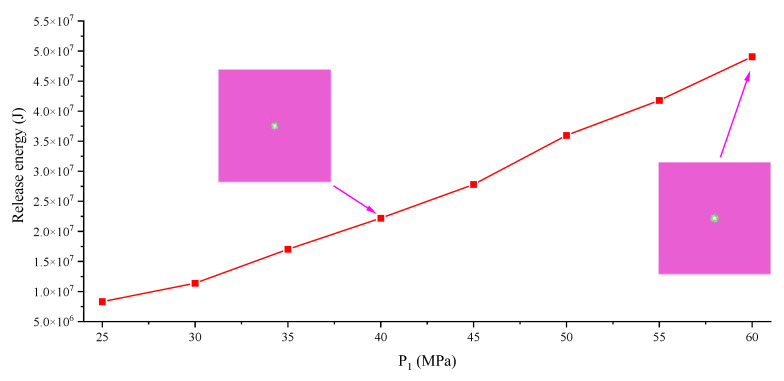
Change curve of release energy under uniform stress field.

**Figure 7 ijerph-19-16636-f007:**
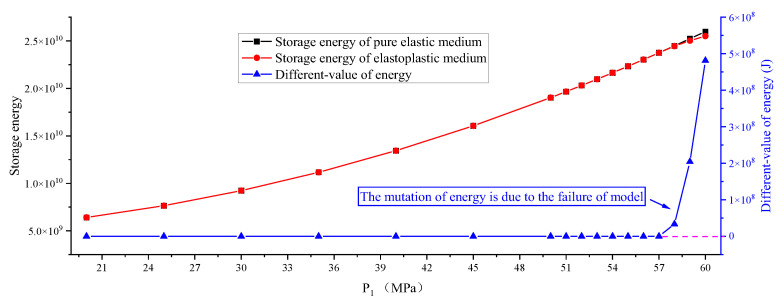
Change curve of storage and different-value of the no-hole model.

**Figure 8 ijerph-19-16636-f008:**
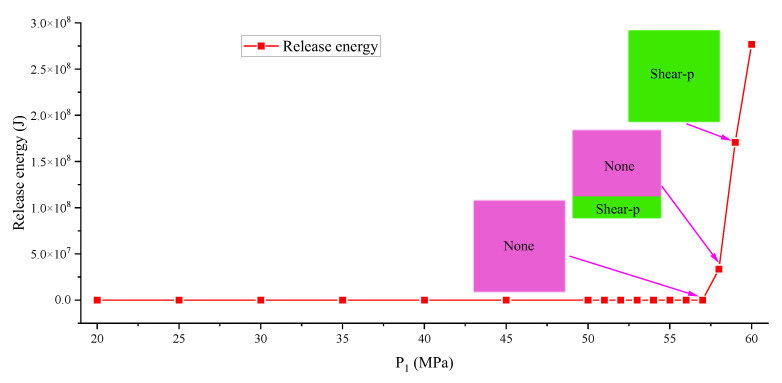
Change curve of release energy of the no-hole model.

**Figure 9 ijerph-19-16636-f009:**
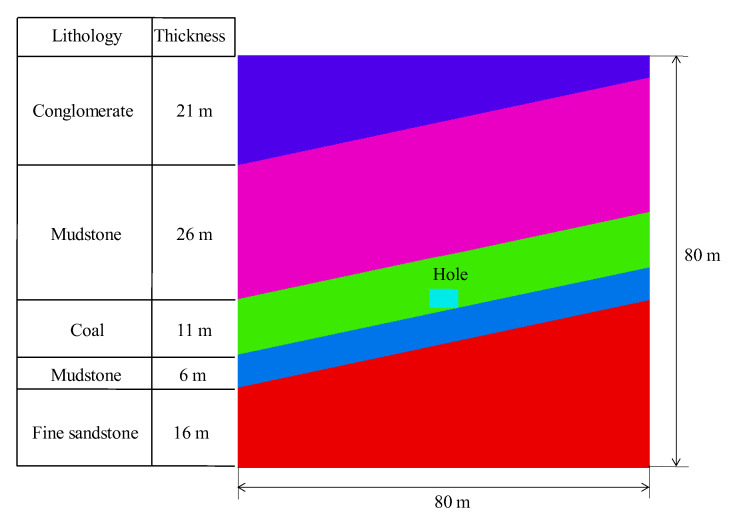
Numerical simulation model of a non-uniform medium.

**Figure 10 ijerph-19-16636-f010:**
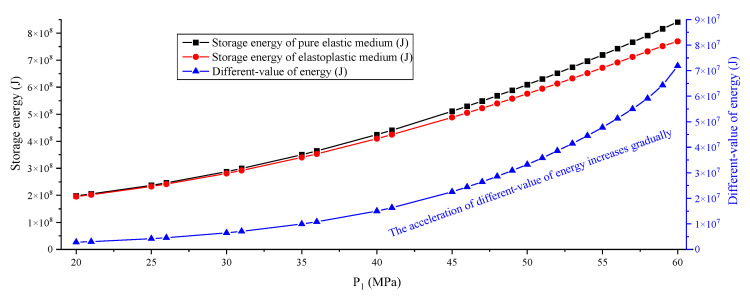
Change curve of storage and different-value of a non-uniform medium.

**Figure 11 ijerph-19-16636-f011:**
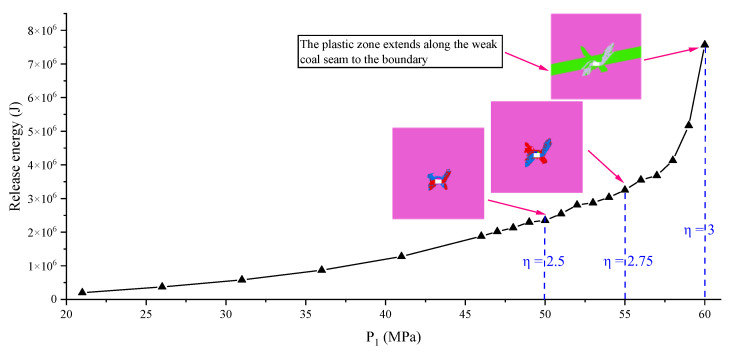
Change curve of release energy of a non-uniform medium.

## Data Availability

Not applicable.

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
