# Peer review of "The Revealed Mechanism of Rock Burst Based on an Innovative Calculation Method of Rock Mass Released Energy"

_ijerph, 2022, doi:10.3390/ijerph192416636_

Round 1

Reviewer 1 Report

In the present work, the authors aimed to calculate the energy released by rock burst to reflect the mechanism, the conditions, and the intensity of this risk. After reviewing the article, the following are my comments.

- The topic is interesting and the presentation of the problem is apparent. In addition, it is well-referenced work and the results are sufficiently discussed. The methodology is simplified and clear, however, the parts describing the comparison cases (hollow/without hollow model, field comparison, uniform/nonuniform medium) in the results section are not clearly stated in the methodology section. Therefore, the methodology section should be improved to include those details.

- Two cases of the hole and no hole results are discussed in this study, why the authors used the hole of 5.6 m? While different hole sizes would change the results, why this factor was not stated?

- The results are released from simulation tools, do the authors agree with the need for validation and improving the relationship with the real-world case? In addition, as this was not stated apparently, please indicate it in the conclusion as a limitation.

- Again, a special material (coal) was used in the uniform model. The need for material justification should be detailed in the methodology section.

- A paragraph of paper organization is suggested to be added to the introduction, as well as, the indication of the limitations and the future research recommendations to be added to the conclusions section. 

- Finally, a final revision is advised to avoid some organization errors (e.g., in lines 106-107: ....established above is shown in Figure 1, whereas it is below) and consistency (e.g., Figure 1 vs Fig. 4 forms). 

Author Response

Reply to Reviewer 1:Thank the reviewer for your excellent comments. Your comments are very helpful to our paper. Here are some specific modifications:

1. The reviewer is quite right. Our research is divided into hole and no hole for comparison. Forgive us for not making it clear. In fact, 5.6 m represents the specific diameter of a field test in a roadway, which is applied to the research model. The diameter of the hole will certainly have an impact on the model energy results. The larger the diameter, the greater the impact. However, this is not the focus of our research. We are already working on the influence of hole diameter on the sample, which is estimated to be reflected in the next article.We added the content in the revised manuscript: “5.6 m represents the specific diameter of a field test in a roadway, which is applied to the research model” and “and the guess is that the larger the diameter, the greater the impact, which will be studied in detail in the later study”

2. The authors fully agree with the need for validation and improving the relationship with the real-world case. So, we added the following content in the conclusion: “However, it should be pointed out that this study relies more on numerical simulation, and later research should focus more on laboratory or field verification.”

3. Thanks for the reminder of the reviewer. The reviewer was right, so we added the content in the revised manuscript: “(rock burst accidents occur more often in coal, so selecting coal as the medium can better reflect the real mechanism)”.

4. We have enriched the last paragraph of the introduction chapter according to your comments. See the revised manuscript for details.

5. Thank the reviewers for their excellent comments. We have unified the relevant statements. See the revised manuscript for details.

Reviewer 2 Report

The study of rock-bursting mechanisms is essential for the development of geotechnical engineering, especially for some underground projects. The authors propose a method for calculating energy release from rock masses and carry out the corresponding study with more rigorous logic and thorough analysis. I think it can be accepted with a minor revision.

1. It should be possible to show more clearly the specific application and implementation of the model.

2. When analyzing the cloud map, the part of the plastic zone should be indicated.

Author Response

Reply to Reviewer 2:

Thank the reviewer for your excellent comments. Your comments are very helpful to our paper. Here are some specific modifications:

  1. The reviewer's comments are very good. We really should show more clearly the specific application and implementation of the model. So, we added the following content in the Numerical methodology chapter: “The obtained model and calculation method quantify the capacity value before and after the occurrence of rock burst, which is of great help to the study of rock burst mechanism in this study. In addition, it can also be used to calculate the damage or accumulation of rock mass energy.”
  2. Thanks for the reminder of the reviewer, we added the following content to the description of the cloud map: “It can also be concluded from the energy difference value distribution that the distribution result of energy is obviously related to the distribution result of plastic zone, and the area producing plastic zone seems to have a greater energy difference.”

Reviewer 3 Report

In the process of deep resource development, rock burst disaster of hard roof with high stress is easily induced due to the influence of geological structure engineering disturbance, so relevant experimental and theoretical research is very important. In this paper, numerical simulation studies are carried out from the perspective of energy and the range of plastic zone of surrounding rock. These works have certain significance, but there are still some minor issues need to be improved:

1: There are many professional grammatical problems in this manuscript, which require the author to carefully revise the whole text: for example, "cohesion" should be "cohesive stress"., etc..

2: In numerical simulation, mechanical parameters of surrounding rock are not introduced in detail. For example, what is the compressive strength? Which plasticity criterion is used?

3: There have been many calculations about the plastic zone of surrounding rock. What is the innovation of this work? In addition, the actual engineering rock stress presents a three-dimensional anisotropic stress state. Does this paper consider the effect of the actual true triaxial stress state? How is the yield constitutive considered?

[1] Xiaofei Guo, Zhiqiang Zhao, Xu Gao, Xiangye Wu, Nianjie Ma. Analytical solutions for characteristic radii of circular roadway surrounding rock plastic zone and their application. International Journal of Mining Science and Technology, 29, 2,2019: 263-272.

[2] Lu J, Yin G, Zhang D, Li X, Huang G, Gao H (2021) Mechanical properties and failure mode of sandstone specimen with a prefabricated borehole under true triaxial stress condition. Geomech Energy Environ 25:100207.

4: What is the direct relationship between the intensity and range of rockburst and the plastic zone of surrounding rock proposed in this paper? Is the rockburst range consistent with the plastic zone range? In addition, how to calculate the stress in the plastic zone of surrounding rock? Is it based on the elastic relation? Is it reasonable?

Author Response

Reply to Reviewer 3:

Thank the reviewer for your excellent comments. Your comments are very helpful to our paper. Here are some specific modifications:

  1. Forgive our carelessness. We read the full text and revised it. See the revised manuscript for details.
  2.  Thanks for the reminder of the reviewer, we added the compressive strength of coal to 15 MPa in the revised manuscript, and explained that the mohr coulomb criterion is used as the failure criterion in the elastoplastic model.
  3. Thank the reviewer for your excellent comments, and forgive us for our unclear expression. In fact, the main innovation of our research is the calculation method of the energy release of rock burst and the mechanism revealed thereby. From our results, we can see that the energy release of the deviatoric stress field is more consistent with the actual accident energy. Other members of our team discussed the selection of the criteria and showed that they are applicable to various criteria.
  4. Thank the reviewer for your excellent comments. According to the research results, when the degree of deviatoric stress field is greater, the area of plastic zone will be larger, the expansion of butterfly leaves will be wider, and the released energy caused by small disturbances will be greater. In the energy calculation model, the energy calculation parameters of the element in the plastic zone adopt the results derived from FLAC3D, and adopt the energy calculation method equivalent to the elastic element, which will bring some errors, but will not change the law of energy release.It is very necessary to study the mechanism of rock burst, which is related to the safe construction of many geotechnical projects. Previous studies have shown that small trigger stress will lead to large energy release, but the specific conditions cause the release and how to quantify the energy are urgent problems to be solved. In this study, an innovative calculation method of rock mass energy release is proposed, and the calculated release energy is consistent with the monitoring results of field monitoring equipment. The revealed mechanism of rock burst reflected is that under the condition of large ratio pre state stress field (mostly>2.5), small trigger stress field will lead to a large amount of energy release under "late butterfly shape" or "final butterfly shape" of the plastic zone. This study reveals the key factor of rock burst, which plays an important reference role for the mechanism research, subsequent monitoring and treatment method of rock burst.

Round 2

Reviewer 3 Report

The current version is can be accepted